# Recent Progress in the Development of Fluorescent Probes for Thiophenol

**DOI:** 10.3390/molecules24203716

**Published:** 2019-10-16

**Authors:** Yuanqiang Hao, Qianye Yin, Yintang Zhang, Maotian Xu, Shu Chen

**Affiliations:** 1Henan Key Laboratory of Biomolecular Recognition and Sensing, College of Chemistry and Chemical Engineering, Henan Joint International Research Laboratory of Chemo/Biosensing and Early Diagnosis of Major Diseases, Shangqiu Normal University, Shangqiu 476000, China; zhangyintang@sqnu.edu.cn (Y.Z.); xumaotian@sqnu.edu.cn (M.X.); 2Key Laboratory of Theoretical Organic Chemistry and Function Molecule of Ministry of Education, School of Chemistry and Chemical Engineering, Hunan University of Science and Technology, Xiangtan 411201, China; yinqy083@nenu.edu.cn; 3College of Chemistry and Molecular Engineering, Zhengzhou University, Zhengzhou 450001, China

**Keywords:** fluorescent probes, thiophenol, review

## Abstract

Thiophenol (PhSH) belongs to a class of highly reactive and toxic aromatic thiols with widespread applications in the chemical industry for preparing pesticides, polymers, and pharmaceuticals. In this review, we comprehensively summarize recent progress in the development of fluorescent probes for detecting and imaging PhSH. These probes are classified according to recognition moieties and are detailed on the basis of their structures and sensing performances. In addition, prospects for future research are also discussed.

## 1. Introduction

Thiophenol (PhSH) and its derivatives are highly reactive aromatic thiols which care extensively used in the chemical industry for preparing pesticides, polymers, and pharmaceuticals [1,2,3]. Despite its critical and widespread usage, PhSH is also highly toxic to organisms. Animal model studies revealed a median lethal concentration of 0.01–0.4 mM for fish and a median lethal dose of 6.2 mg/kg in mouse [4,5]. Exposure to PhSH can cause serious health problems to the human body including shortness of breath, muscular weakness, kidney and liver malfunctions, central nervous injuries, and even death. Thus, it is of great importance to develop highly sensitive and selective methods for monitoring PhSH in environmental and biological systems. 

Analytical assay-based synthetic fluorescent probes are highly attractive and versatile for the detection of biologically and/or environmentally important species because of their high selectivity sensitive, ease of operation, tunable photophysical properties, etc. [6,7,8,9,10,11,12]. In recent years, significant progress was made in this field, and numerous effective fluorescent probes were developed for sensing and imaging applications [13,14,15,16,17,18,19,20,21,22,23,24,25,26,27,28,29]. Thiols (including biothiol, hydrogen sulfide, and thiophenol) are important compounds with some unique properties such as strong nucleophilicity, coordination ability, and reducing capacity [30,31,32,33,34,35,36,37,38,39,40,41,42]. By exploiting these properties, a number of fluorescent probes were reported for sensing thiols [43,44,45,46]. Among these probes, 2,4-dinitrobenzenesulfonyl (DNPS) and 2,4-dinitrophenyl (DNP) are two commonly used recognition moieties, which can act as favored sites for nucleophilic attack by thiols [47,48,49,50]. Compared with biothiols, such as glutathione (GSH), cysteine (Cys), and homocysteine (Hcy), PhSH has a higher pK_a_ value (ca. 6.5 for PhSH and ca. 8.5 for biothiols); thus, it mostly (about 98%) exists as the anionic PhS^−^ under neutral physiological conditions (pH = 7.4) while biothiols remain as the neutral form (RSH). Consequently, PhSH can act as a more powerful nucleophile compared with biothiols. On the basis of the different reactivities, a number of selective PhSH probes were obtained by incorporating the DNPS or DNP moiety into various screened fluorophores. Although rapid progress was achieved in recent years in developing fluorescent assays for sensing PhSH based on synthetic fluorescent probes, these studies are yet to be comprehensively addressed. In this review, the reports of fluorescent PhSH probes (including several colorimetric and luminescent probes) are systematically summarized and classified according to recognition moieties. Prospects for future research are discussed as well.

## 2. Probes Based on 2,4-Dinitrobenzenesulfonic Amide

Wang et. al. reported the first fluorescent probe (**1**) for the selective detection of PhSH [51]. The probe was obtained by incorporating DNBS into an NBD (7-nitro-2,1,3-benzoxadiazole) fluorophore (Figure 1). Due to the blockage of the ICT (intramolecular charge transfer) process and the quenching effects of the DNBS moiety, probe **1** was non-fluorescent. In an aqueous buffer solution, probe **1** can display a fast turn-on fluorescence response due to the effective cleavage of the sulfonamide by PhSH via an S_N_Ar (nucleophilic aromatic substitution) process and, thus, release the free fluorophore NBD-NH_2_. Notably, the probe demonstrates excellent selectivity over aliphatic thiols and other common nucleophiles. Considering the relatively low quantum yield (Φ = 0.02) and low sensitivity (limit of detection, LOD = 2 μM) of the probe **1** system, the same research group subsequently developed a novel PET (photoinduced electron transfer)-based fluorescent probe (**2**) for sensing PhSH [52], which displayed a much higher quantum yield (Φ = 0.39; quantum yield was determined by reference to harmine in 0.1 N H_2_SO_4_ (Φ’ = 0.45)) and a higher sensitivity (LOD = 0.2 μM). Probe **2** also displayed excellent specificity and can work in a phosphate buffer under physiological conditions (pH 7.3, 0.01 M).

Since the pioneering works by Wang and co-workers, considerable efforts were expended for the development of efficient fluorescent PhSH probes. By appending DNBS to a naphthalimide-derived fluorophore, Han and Deng et al. designed a reaction-based fluorescent, probe (**3**) (Figure 2) for PhSH [53]. Probe **3** was shown to be selective and sensitive (LOD = 20 nM), which was presumably ascribed to the suitable electronic structure and spectroscopic property of the probe, benefiting from the linker moiety of 2,3-dihydroimidazo-[1,2-a] pyridine. In the presence of PhSH (ca. 2.0 equivalents), the probe solution showed a remarkable turn-on fluorescent response (>60-fold) with a fluorescence quantum yield of 0.36. Furthermore, probe **3** was used successfully for monitoring PhSH in water samples from rivers and chemical factories; the feasibility of the proposed assay was also validated using the traditional GC–FPD (gas chromatography–flame photometric detector) method. 

By adopting coumarin-3-amino as the fluorophore, Yang et al. developed a fluorescent PhSH probe (**4**) (Figure 3) [54]. In a phosphate buffer solution, probe **4** displayed a significant fluorescence enhancement (>280-fold) with a large Stokes shift (145 nm) for sensing PhSH. Probe **4** also exhibited excellent selectivity for PhSH over relevant aliphatic thiols. Thus, the probe was directly applied to detection PhSH in real water samples, which produced good recovery results, indicating that probe **4** can serve as a promising tool for sensing thiophenol in the environmental field. To further improve the analytical performances of the coumarin-based fluorescent probe, the same research group designed a new probe (**5**) for PhSH via a twist-blockage strategy [55]. The benzo[ij]quinolizine coumarin-based probe (**5**) exhibited more advantageous features, including higher fluorescence enhancement (700-fold) and sensitivity (LOD = 4.5 nM). Probe **5** was also employed for imaging exogenous PhSH in HEK293 cells.

Boron dipyrromethene difluoride (BODIPY)-based dyes have several advantages such as a high molar extinction coefficient and fluorescence quantum yield, excellent photostability, and narrow emission bands [56,57,58,59]. Talukdar et al. reported a BODIPY-based probe (**6**) (Figure 4) for fluorescence turn-on detection of PhSH [60]. The fluorescence off–on process was confirmed by theoretical calculations. Probe **6** displayed a forbidden S_0_/S_1_ transition (oscillator strength, *f* = 0.0561), while the presumed reaction product of **6** with PhSH allowed the S_0_/S_1_ transition (*f* = 0.4693) (Figure 6c). In phosphate buffer (containing 1% dimethyl sulfoxide (DMSO), pH 7.3), the fluorescence intensity of the probe **6** system was found to increase linearly with PhSH concentration. The dynamic range was 2–10 μM with an LOD of 34.4 nM. By employing an extended π-conjugated BODIPY dye, Zhang and Zhao et al. developed a long-wavelength PhSH probe (**7**) (Figure 4) with the emission band located around 640 nm [61]. Based on an amino phenothiazine boranil dye, Chen and Sheng et al. developed a highly sensitive fluorescent probe (**8**) (Figure 4) for sensing PhSH [62]. Recently, Thilagar et al. synthesized two triarylborane-derived fluorescent PhSH probes (**9** and **10**) (Figure 4), which displayed high sensitivity and selectivity, as well as applicability for intracellular imaging [63]. 

Luminescent heavy-metal complexes showed some favorable photophysical characteristics for sensing and bioimaging applications, such as a large Stokes shift and long emission lifetime, which are very helpful for eliminating autofluorescence in complicated biological and environmental samples [64,65,66,67,68,69,70]. Lu et al. presented a cyclometalated iridium-based phosphorescent probe (**11**) (Figure 5) for sensing PhSH [71]. Probe **11** is almost non-fluorescent due to the PET quenching process. PhSH can trigger a distinct turn-on photoluminescence for the probe system. The photoluminescence lifetimes were determined to be 1.99 μs for de-aerated and 0.50 μs for aerated samples. The photoluminescence response was also very sensitive (*k =* 91.92 × 10^6^ M^−1^) with a determined LOD of 2.5 nM. Recently, Zhang and Pu et al. reported another cyclometalated iridium(III) complex (**12**) (Figure 5) for detecting PhSH [72]. 

Near-infrared (NIR) fluorescent probes attracted great attention due to their better capability for deep tissue penetration and reducing phototoxicity [73,74,75]. Up to now, several near-infrared probes based on the recognition unit of 2,4-dinitrobenzenesulfonic amide were reported for sensing PhSH. Based on a dicyanomethylene–benzopyran-derived dye, Feng et al. developed an NIR fluorescent probe (**13**) (Figure 6a) for sensing and imaging PhSH [76]. Upon addition of PhSH, the probe buffer solution displayed a distinct turn-on NIR fluorescence around 670 nm. The probe system showed large Stokes shift, high selectivity, and fast response time. The linear dynamic range was 1–10 μM with an LOD of 0.15 μM. Furthermore, probe **13** was demonstrated to be capable of imaging PhSH in living Hela cells (Figure 6b–e). Subsequently, three other NIR fluorescent PhSH probes (**14** [77], **15** [78,79], and **16** [79]) emerged by adopting different fluorophores, including porphyrin, dicyanoisophorone, and Nile blue (Figure 7). By incorporating 2,4-dinitrobenzenesulfonic amide into other fluorophores, a number of fluorescent PhSH probes (**17** [80], **18** [80], **19** [81], **20** [82], and **21** [83]) were also constructed (Figure 8). 

Recently, by introducing piperazine as a linker for connecting the fluorophore and the recognition moiety, a series of new fluorescent probes were developed for monitoring PhSH. Yu and Li et al. reported a 1,8-naphthalimide-based fluorescent PhSH probe (**22**) (Figure 9) [84]. In a mixed solvent of water/ethanol (7:3, *v*/*v*), the probe displayed a turn-on fluorescence response toward PhSH, which can be ascribed to target-induced cleavage of the 2,4-dinitrobenzenesulfonyl group and, thus, the release of the piperazine-appended 1,8-naphthalimide fluorophore. This sensing process was verified by NMR (nuclear magnetic resonance) and MS (mass spectroscopy) studies. Probe **22** was also shown to be very sensitive for PhSH, as the LOD was determined to be 10.3 nM. Based on the similar designing strategy, several other fluorescent probes (**23** [85], **24** [86], **25** [79], and **26** [87]) bearing a piperazine linker were also built for the detection of PhSH (Figure 10). Notably, probe **26**, which was developed by He and Chen et al., consisted of two types of fluorophores (naphthalimide and dansyl) linked by the piperazine group. The dansyl group behaves not only as a recognition moiety but also as a signaling reporter. Probe **26** alone displayed a typical naphthalimide emission at 534 nm, while a blue-shifted peak at 414 nm was gradually observed in the presence of PhSH, which can be attributed to the release of the dansyl derivative. Thus, the ratiometric detection for PhSH was achieved. 

Photochromic molecules, which can perform a reversible phototransformation between two forms that have different absorption spectra, showed many promising applications in molecular switches, photonic devices, and sensors. Based on a photochromic diarylethene derivative, Cheng and Zhang et al. developed a colorimetric PhSH probe (**27**) (Figure 11) [88]. Probe **27**, which was obtained by appending 2,4-dinitrobenzenesulfonyl groups to the bisthienylethene perfluordiarylethene, shows excellent reversible photochromic properties. It can be photoswitched between a colorless open form and a colored closed form upon irradiation with ultraviolet (UV) light/visible light. After reacting with PhSH, the photochromism of the probe system can be effectively inhibited, with the photocyclization quantum yields (Φ_o-c_) decreasing from 0.25 to 0.01.

## 3. Probes Based on 2,4-Dinitrobenzenesulfonate

By using 2,4-dinitrobenzene-sulfonate as the recognition moiety, Zhang et al. reported the first two-photon fluorescent probe (**28**) (Figure 12) for PhSH detection [89]. The signal reporter of this probe is a TBET (through-bond energy transfer) system which was obtained by combining a donor of two-photon naphthalene with an acceptor of red-emissive BODIPY dye. The TBET fluorophore displayed a high energy transfer efficiency (93.5%) from the donor to the acceptor. In the presence of PhSH, the probe exhibited a high-sensitivity turn-on fluorescence response of the BODIPY acceptor at 586 nm. The linear range of the PhSH was 0.05–2 μM, and the LOD corresponded to 4.9 nM. Furthermore, probe **28** was utilized for thymidine phosphorylase (TP) bioimaging PhSH in Hela cells, as well as in rat liver tissue slices.

Squaraines, which possess a donor–acceptor–donor (D-A-D′) and resonance-stabilized zwitterionic structure, exhibited several unique photophysical features for the construction of fluorescent probes, such as narrow and strong absorption and emission bands in the red to NIR region, excellent photostability, and tunable fluorescence emission properties [90,91,92]. Based on an arylidene–squaraine fluorophore, Lu et al. developed a far-red/near-infrared fluorescent probe (**29**) (Figure 13) for sensing PhSH [93]. Probe **29** exhibited both a sensitive colorimetric response (from pink to blue) and a fluorogenic response toward PhSH. Due to the desirable electronic and spatial structures of the employed donating building blocks (D: benzindolylmethylene group and D′: 2,4,6-trihydroxyphenyl group) connected to the squaraine core, the cyclobutene ring displayed good stability against nucleophilic attack by biothiols, such as GSH and Cys. Thus, the probe exhibited good selectivity. The probe was also very sensitive toward PhSH; the dynamic range and LOD were 0–5 μM and 9.9 nM, respectively. 

By incorporating 2,4-dinitrobenzene-sulfonate into an aldehyded dicyanomethylene–benzopyran-derived dye, Li et al. constructed near-infrared probe (**30**) (Figure 14) for colorimetric and fluorogenic detection of PhSH [94]. With the addition of PhSH to a phosphate buffer/DMSO (*v*/*v*, 1:1, pH = 7.4) solution of probe **30**, the probe solution displayed a distinct color change from yellow to red with the evolution of a fluorescence emission at 658 nm. The probe displayed excellent selectivity for PhSH amongst a selection of nucleophiles, including aniline, Cys, Hcy, and GSH. The LOD was calculated to be 0.22 μM. The probe was also employed for sensing PhSH in tap water and in river water samples, as well as imaging the target in living Hela cells. 

By exploiting natural curcumin as the fluorophore, Martínez-Máñez and Yin et al. developed a turn-on fluorescent PhSH probe (**31**) (Figure 15) [95]. In a mixed solvent of HEPES (*N*-2-hydroxyethylpiperazine-*N*-2′-ethanesulfonic acid)–MeOH (methanol) (1:1, *v*/*v*, pH = 7.4), PhSH can rapidly react with probe **31** and produce a significant fluorescence enhancement (20-fold) at around 536 nm. The reaction was complete within dozens of seconds. The probe was also used for detection of PhSH in HepG2 cells. Combining 2,4-dinitrobenzene-sulfonate with various other fluorophores, including coumarin, benzoxazinone, naphthalene, and BODIPY, further yielded a number of fluorescent PhSH probes (**31** [95], **32** [96], **33** [97], **34** [98], **35** [99], **36** [100], **37** [101], **38** [102], and **39** [103]) (Figure 15).

## 4. Probes Based on the Recognition Unit of 2,4-Dinitrophenyl Ether

Lin et al. firstly proposed constructing a fluorescent PhSH probe (**40**) (Figure 16) by exploiting the target-induced thiolysis of a dinitrophenyl ether moiety [104]. The probe (**40**) was obtained by coupling a dinitrophenyl group to a 7-hydroxy coumarin. In a solution of phosphate buffer/DMF (*N*,*N*-dimethylformamide) (55:45, *v*/*v*, 25 mM, pH = 7.0), the probe is almost non-fluorescent (Φ = 0.006), while PhSH can effectively mediate thiolysis of dinitrophenyl ether and result in the release of the highly emissive coumarin fluorophore (Φ = 0.5). The quantum yield was calculated using the equation of Φ_s_ = Φ_r_ (A_r_F_s_/A_s_F_r_) (n_s_/n_r_)^2^, where A_s_ and A_r_ are the absorbance of the sample and the reference at the excitation wavelength, respectively, F_s_ and F_r_ are the corresponding relative integrated fluorescence intensities, n is the refractive index of the solvent, and Φ_s_ and Φ_r_ are the quantum yields of the sample and the reference, respectively; quinine sulfate (Φ_r_ = 0.546 in 1 N H_2_SO_4_) was used as the reference. The kinetic results indicated that the probe can react rapidly with the target. The spectral response of the probe solution was found to be very sensitive toward PhSH. A linear dynamic range of 4.0 nM–3.0 μM was obtained between the logarithm of ((*I* − *I*_0_)/*I*_0_), and the LOD was calculated to be 1.8 nM. Moreover, the probe was used for the first time to monitor PhSH in environmental and biological samples. 

Compared with the intensity-dependent single-band emission probe, the ratiometric fluorescent probe, which relies on the change in the ratio of two emission bands, is more attractive due to its potential capability to eliminate the influence of probe distribution and instrumental difference. The fluorescence resonance energy transfer (FRET) system, which consists of two dyes (donor and acceptor), is a commonly used platform for constructing ratiometric fluorescent assays [105,106,107,108]. Feng et al. developed a FRET-based ratiometric PhSH probe (**41**) (Figure 17) by using a pair of fluorophores, coumarin and naphthalimide [109]. As the fluorescence of the naphthalimide acceptor was quenched by dinitrophenyl ether, probe **41** alone displayed the typical emission (λ_em_ = 481 nm) of the coumarin donor. PhSH can selectively remove the dinitrophenyl ether and trigger the occurrence of the FRET process, thus achieving a good ratiometric fluorescent response. 

Based on the strategy of target-induced in situ formation of a red-emissive coumarin fluorophore, Song et al. reported a reaction-based turn-on fluorescent probe (**42**) (Figure 18) for sensing PhSH [110]. Firstly, the probe underwent a target-mediated cleavage of the dinitrophenyl ether moiety in the presence of PhSH. Then, the liberated phenolic oxygen could subsequently perform an intramolecular nucleophilic attack on the nitrile group, leading to the generation of the cyclized iminocoumarin scaffold acting as the signal reporter. The sensing process was further verified by HPLC (high-performance liquid chromatography), NMR, and MS analysis. The probe also displayed a very fast reaction rate (within 120 s) for response to PhSH. Exploiting a similar sensing strategy, Chen and Sheng et al. consequently reported a phenothiazine–coumarin-based red-emitting probe (**43**) for detection of PhSH [111]. In aqueous solution, the formed imino group via PhSH-induced cleavage and an intramolecular cyclization process could further be converted to oxygen via a hydrolysis reaction. Probe **43** exhibited an unusually large Stokes shift (140 nm) and a high sensitivity (LOD = 2.9 nM) for sensing PhSH. 

By integrating different fluorophores and reaction sites into a single molecular probe, Song et al. established a potent sensing platform (probe **44**) for differentiating various thiols (including Cys/Hcy, GSH/H_2_S, and thiophenol) [112]. Probe **44** incorporated three latent fluorometric fluorophores, blue-emitting coumarin, green-emitting NBD, and a red-emitting coumarin analogue. As shown in Figure 19a, Cys/Hcy, GSH/H_2_S, and thiophenol can generate different combinations of fluorescence signals, corresponding to blue–green, blue, and blue–red, respectively. Moreover, this probe system also can be utilized for discriminating these groups of thiols (Cys/Hcy, GSH/H_2_S, and thiophenol) in living cells via recording fluorescence signals in different spectral channels (Figure 19b). 

Excited-state intramolecular proton transfer (ESIPT)-based fluorescent probes are highly attractive due to their unique optical features, such as large Stokes shift, high fluorescence quantum yield, tunable structure and emission wavelength, and potential for ratiometric sensing [8,113]. By coupling 2,4-dinitrophenyl ether to a green-emissive ESIPT fluorophore, 3-hydroxyphthalimide, Song et al. synthesized a turn-on fluorescent PhSH probe (**45**) (Figure 20) [114]. Due to the PET process from the fluorophore to the dinitrophenyl ether moiety, probe **45** is non-fluorescent (Φ_F_ < 0.001). Upon addition of PhSH, the dinitrophenyl ether group of the probe can be cleaved to liberate 3-hydroxyphthalimide which can undergo an ESIPT process under photoexcitation and exhibit the keto-form emission at 516 nm. The probe system displayed a large Stokes shift (161 nm). The dynamic range and LOD for sensing PhSH are 0–10 μM and 3.5 nM, respectively. By using other ESIPT fluorophores, including 2-(2′-hydroxy-phenyl)benzothiazole (HBT), an imidazo [1,5-α]pyridine derivative, and phenothiazine-HBT compound, three other fluorescent PhSH probes (**46** [115,116], **47** [117], and **48** [118]) were also presented (Figure 20). 

Positively charged dyes showed great potential for the construction of fluorescent probes, due to their favorable electronic structures and optical properties, improved water solubility, and mitochondrial targeting ability [119,120,121]. By using 7-hydroxy quinolinium as the fluorophore, Song et al. developed an off–on fluorescent probe (**49**) (Figure 21) for sensing PhSH [122]. The probe can quickly respond to PhSH with a rate constant of *k* = 24 M^−1^∙s^−1^. Studies on pH effects revealed that the probe can effectively react with PhSH under neutral physiological conditions. Liu and Song et al. reported a red-emissive fluorescent PhSH probe (**50**) (Figure 21) based on a methylated chromenoquinoline dye [123]. The probe system displayed a large Stokes shift (108 nm) and a high sensitivity (LOD = 8.1 nM) for sensing PhSH. Kinetic studies revealed that the reaction with probe **50** can be completed within 2 min. The probe also showed excellent specificity for PhSH as other nucleophilic reagents (including Cys, Hcy, GSH, and HS^−^) did not generate any fluorescence response. Guo et al. presented a mitochondrial-targeted fluorescent PhSH probe (**51**) (Figure 21) by using a rhodol–methylpyridinium derivative as the fluorophore [124]. Probe **51** can display both a distinct colorimetric and a fluorogenic response toward PhSH in a solvent of phosphate buffer (pH = 7.4, 20 mM, containing 20% DMF). Cellular co-localization experiments showed that the probe can stain mitochondria with a high coefficient (0.95) using Mito Tracker Green FM as the reference. The effects of endogenous reactive oxygen species (ROS) on the fluorescence response of probe **51** to PhSH in living cells were also inspected, which revealed that endogenous ROS could eliminate the fluorescence of the probe system. The results implied that thiophenols can participate in oxidative stress and redox regulation reactions. Zeng and Yuan et al. developed a selective NIR fluorescent probe (**52**) (Figure 21) for sensing and imaging PhSH by screening a library of seven fluorescent probes which were established by installing nitrobenzenesulfonyl or nitrophenyl groups onto NIR chromenylium–cyanine fluorophores [125]. The probe has both excitation and emission in the NIR region, corresponding to 680 and 706 nm, respectively. In a phosphate buffer solution (25 mM, pH 7.4, containing 1% DMSO), the emitted fluorescence intensity of probe **51** at 706 nm increased linearly with concentration of PhSH in a range of 1–15 μM with an LOD of 280 nM. Furthermore, the probe exhibited excellent biocompatibility and cell-membrane permeability, and was successfully exploited for imaging PhSH in living HeLa cells, as well as in living mice. 

The emissive ruthenium complex is a promising luminescent material for sensing and imaging applications due to its unique photophysical properties, such as high molar extinction coefficient, high luminescent efficiency, large Stokes shift, long luminescence lifetime, and enhanced thermal and photo-chemical stabilities [126,127,128,129,130]. By utilizing 2,2’-bipyridine (bpy) and 4-(4-(2,4-dinitrophenoxy)phenyl)-2,2’-bipyridine (DNP-bpy) as ligands, Yuan et al. synthesized a series of Ru derivatives (**53**–**55**) (Figure 22a), [Ru(bpy)_3__-n_(DNP-bpy)_n_](PF_6_)_2_ (n = 1, 2, 3), for luminescent sensing PhSH [131]. Due to the PET process from the luminescent Ru center to the 2,4-dinitrophenyl moiety, these probes are only weakly emissive. The luminescent behaviors of probes **53**–**55** and their responses toward PhSH were systematically studied. Among these probes, [Ru(bpy)(DNP-bpy)_2_]^2+^ (**54**) had the lowest quantum yield and displayed the most remarkable turn-on luminescent response to PhSH. The results also demonstrate that the symmetry of the ligands is a critical factor influencing the luminescence features of the Ru-complex. In an aqueous buffer solution (20 mM HEPES, pH 7.0), probe **54** could sensitively respond to PhSH with a linear dynamic range of 1–12 μM and an LOD of 32.9 nM. Moreover, all these probes showed high specificity toward PhSH. Probe **54** can also be exploited for imaging exogenous PhSH in living cells (Figure 22b). Further cellular imaging experiments demonstrated that probe **54** has excellent intracellular retention, long-term stability to resist photobleaching, low cytotoxicity, and the tendency of nuclear localization. 

Chemiluminescence- and bioluminescence-based analytical assays have a number of advantages including simple procedures and equipment as no excitation light is needed, low background signal and high sensitivity, and good biocompatibility [132,133]. Li et al. reported a dual bioluminescent and chemiluminescent PhSH probe (**56**) (Figure 23) based on coelenterazine [134]. In Tris-HCl buffer (50 mM, pH 7.4) with the addition of luciferase, the probe system can be used for bioluminescent sensing of PhSH with an emission peak at 445 nm. When replacing luciferase with dimethyl sulfoxide (DMSO), the probe system can act as a chemiluminescent sensor for PhSH with an emission peak at 522 nm. Both of these luminescent assays displayed a linear range of 0.5–20 µM for PhSH. Moreover, probe **56** can be employed for quantitative chemiluminescent monitoring of PhSH in plasma samples without the addition of luciferase or DMSO. 

Electrochemiluminescence (also referred to as electrogenerated chemiluminescence, ECL) sensors drew a great deal of attention due to their distinct advantages such as high sensitivity and real-time analysis [135]. The commonly used ECL luminophores include transition-metal complexes, polyaromatic hydrocarbons, and semiconductor nanocrystals [136]. Among them, transition-metal complexes (i.e., Ru, Ir, and Os) are particularly attractive because of their wide variety of structures and photophysics properties. Using 1-phenylisoquinoline/4′-formyl-1-phenylisoquinoline and acetylacetone as the ligands, Hong et al. developed two cyclometalated iridium(III) complexes (**57**, **58**) as both photoluminescence (PL) and ECL probes for sensing PhSH [137]. The structures and ECL sensing mechanisms of these probes are shown in Figure 24. Probe **58** contained an electron-withdrawing formyl group on the quinoline ligand and, thus, displayed faster kinetics and higher sensitivity for responding to PhSH compared with probe **57**. The ECL sensing process involved four steps (i.e., probe **58**): (i) PhSH-mediated thiolysis of dinitrophenyl ether and the release of the Ir-based luminophore; (ii) oxidation of probe **58** at Pt electrode to generate **(58-DNP)**^•**+**^; (iii) one-electron transfer from the tripropylamine radical TPrA^•^ to **(58-DNP)**^•**+**^ to afford **(58-DNP)**^*^; (iv) excited-state **(58-DNP)**^*^ emission of light. The sensing mechanism was also confirmed by density functional theory (DFT) calculations via theoretically predicting highest occupied molecular orbital (HOMO) and lowest unoccupied molecular orbital (LUMO) energies of different species**.** In a mixed solvent of aqueous HEPES buffer and CH_3_CN (1:1, *v*/*v*, 10 mM, pH 7.4, containing TprA (tripropylamine) as the co-reactant and TBAP (tetrabutylammonium perchlorate) as the supporting electrolyte), probe **58** displayed a sensitive ECL response to PhSH at 1.4 V. Two linear dynamic ranges were obtained between the ECL intensities and PhSH concentration, 0.5–20 µM (*y* = 0.26*x* − 1.1) and 0–100 nM (*y* = (1.7 × 10^−2^)*x* + 1.0). The LOD was calculated to be 3.8 nM. Competitive analysis revealed that other analytes could not generate any noticeable ECL changes, while remarkable ECL signals could be observed by adding PhSH to mixtures of probe **58** with other interference, except for the oxidation-sensitive anion of iodide. 

By using a well-known pH indicator, phenolphthalein, as the signaling reporter, Park and Yoon et al. reported a colorimetric probe (**59**) (Figure 25) for sensing PhSH [138]. Incorporation of dinitrophenyl ether into a number of other fluorophores afforded PhSH probes with different emission colors (Figure 25 and Figure 26): cyan (**60** [139]), green (**61** [140], **62** [141], **63** [142], **64** [143], **65** [144], **66** [145], and **67** [146]), yellow–green (**68** [147] and **69** [148]), orange (**70** [142]), red (**71** [149], **72** [150], and **73** [151]), and near-infrared (**74** [152], **75** [153], and **76** [154]). 

## 5. Probes Based on Other Recognition Moieties

By utilizing several modified pyrimidine ethers as the recognition moieties and fluorescein/resorufin as the fluorophores, Wang and Tang et al. established a fluorescent probe library for screening selective probes toward certain sulfhydryl compounds [155]. Among them, probe Flu-3 (**77**), which was obtained by reacting fluorescein with 4-chloropyrimidine, exhibited high sensitivity and specificity for sensing PhSH. The structure and sensing mechanism of probe **77** for PhSH is shown in Figure 27. In HEPES buffer solution (20 mM, pH 7.4), the emission of probe **77** centered at 519 nm, corresponding to the emission of fluorescein, was found to be significantly enhanced upon the addition of PhSH. A good linearity between the fluorescence intensity and the concentration of PhSH in the range of 0–100 μM was obtained, and the LOD was determined to be 1.8 μM. Probe 77 was also employed for fluorescence imaging of thiophenol in HepG2 cells. 

Niu and Yang et al. reported a chlorinated BODIPY-based fluorescent probe (**78**) (Figure 28) for ratiometric sensing PhSH [156]. In a solution of acetonitrile/HEPES buffer (1:3 *v*/*v*, 20 mM, pH 7.4), the probe displayed an absorption maximum at 518 nm and a distinct green emission (λ_em_ = 540 nm) upon reacting with PhSH via nucleophilic substitution. The absorption band shifted to 558 nm with the solution color changing from orange to pink (Figure 28b), and the emission band shifted to 580 nm corresponding to a bright-yellow fluorescence (Figure 28c). The ratio of fluorescence intensities at 581 and 540 nm (*I*_581_/*I*_540_) could remarkably increase (up to 640-fold) in the presence of an excess amount of PhSH. A good linear relationship was obtained between the fluorescence ratio and concentration of PhSH in the range of 0–18 μM (*R*^2^ = 0.997). The LOD was determined to be 36.9 nM. The probe also displayed excellent selectivity and anti-interference ability for sensing PhSH. Furthermore, probe **78** was successfully exploited for ratiometric imaging of PhSH in living HeLa cells via observing the intracellular fluorescence in both green and red channels (Figure 28d–k). 

In previous studies, nitrobenzoxadiazole (NBD)-appended dual-fluorophore (NBD-O-F) systems were artfully employed for differentiating H_2_S/GSH and Cys/Hcy (Figure 29a) [157,158,159]. Typically, thiols can readily react with NBD-O-F via nucleophilic aromatic substitution to produce NBD-SR (or NBD-SH for H_2_S) and another free fluorophore (marked as F). In cases of Cys and Hcy, the generated NBD-SR can undergo subsequent intermolecular rearrangement with adjacent amine groups to form emissive NBD–NHR derivatives, while such a process cannot occur for H_2_S and GSH. Recently, Chen and Zeng et al. synthesized an NBD-O-porphyrin (probe **79**) (Figure 29b) by installing NBD onto a porphyrin dye at the aromatic *ortho*-position [160]. Interestingly, probe **79** displayed selective turn-on fluorescence toward PhSH. The markedly suppressed reactivities of the probe for other thiols (Cys, Hcy, GSH) can be ascribed to the increased steric hindrance effect of these biothiols. Probe **79** also showed great sensing performance for PhSH, including fast response (60 s), high sensitivity (LOD = 54 nM), and deep-red to NIR emission. 

## 6. Conclusions

Since Wang et al. reported the first synthetic chemodosimeter for the selective detection of PhSH in 2007, a great deal of effort was devoted to the development of fluorescent PhSH probes. In this review, we comprehensively overviewed the recently reported fluorogenic and colorimetric probes for sensing PhSH. Table 1 shows the structures and analytical performances of some representative probes for sensing PhSH. Notably, luminophores for chemiluminescence sensing of PhSH were also devised, which could provide a new strategy for extending the applications of synthetic organic probes in other advanced analytical techniques. By adapting various fluorophores, the developed fluorescent PhSH probes cover spectral regions from ultraviolet to visible and even near-infrared. Some of these fluorescent probes displayed superior analytical performance, such as high sensitivity, fast response, good specificity, high photo- and chemical stabilities, high quantum yields, etc. Although substantial and rapid progress h was made in the development of effective fluorescent probes for PhSH, the recognition moieties and reaction types of these probes for PhSH are still quite limited. Most of these probes were obtained by directly appending amine- or hydroxyl-containing fluorophores with the recognition unit of 2,4-dinitrobenzenesulfonyl/2,4-dinitrophenyl groups, which have dramatic fluorescence-quenching effects. Upon PhSH-mediated cleavage of the recognition unit, the probe system can restore the fluorescence spectra of the fluorophore; thus, these probes display “turn-on” fluorescence behavior. Future efforts can aim at exploring new appropriate electron-deficient or potential leaving groups as the recognition moieties to sensitively respond to PhSH. Introducing a self-immolative linker between the fluorophore and the recognition unit is also a feasible approach for establishing ratiometric probes.

## Figures and Tables

**Figure 1 molecules-24-03716-f001:**
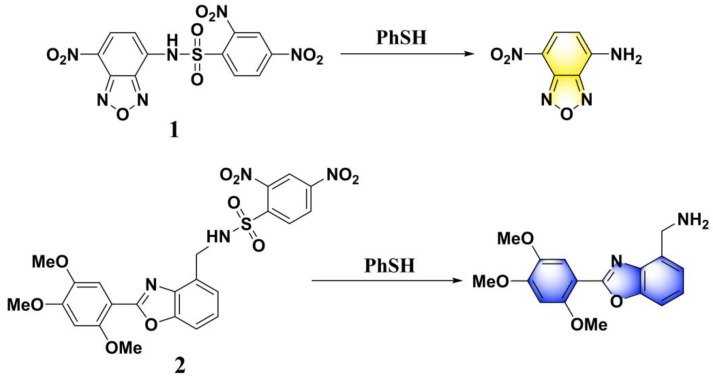
Structures and reactions of probes **1** and **2** with thiophenol (PhSH).

**Figure 2 molecules-24-03716-f002:**
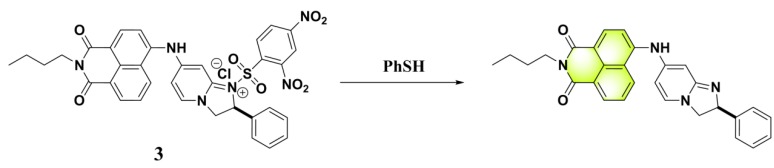
Structure and reaction of probe **3** with PhSH.

**Figure 3 molecules-24-03716-f003:**
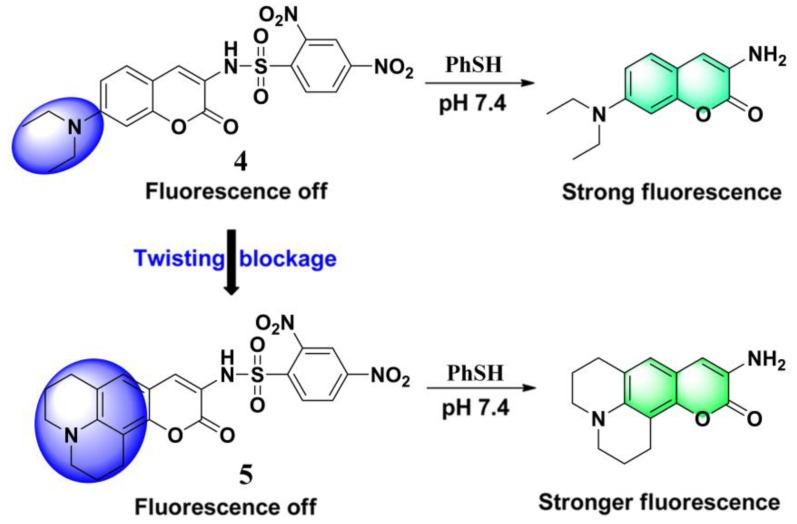
Structures and reactions of probes **4** and **5** with PhSH. Reproduced with permission from References [54,55]; copyright American Chemical Society.

**Figure 4 molecules-24-03716-f004:**
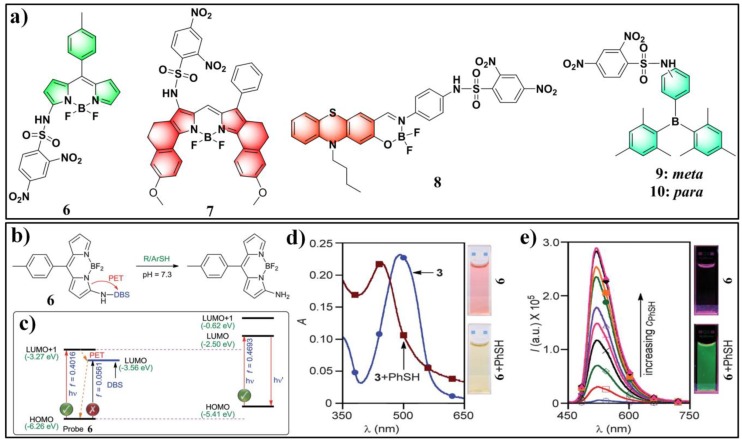
(**a**) Fluorescent PhSH probes (**6**–**10**) based on boron-containing dyes. (**b**) Reaction of probe **6** with PhSH. (**c**) Energy level diagram of the frontier molecular orbitals (MOs) of probe **6** and corresponding amine. (**d**) Ultraviolet–visible light (UV–Vis) absorption spectra of probe **6** before and after addition of PhSH; photographs were taken under ambient light. (**e**) Fluorescence spectra of **6** in the presence of different concentrations of PhSH. Photographs taken under a hand-held UV lamp. Reproduced with permission from Reference [60]; copyright Royal Society of Chemistry.

**Figure 5 molecules-24-03716-f005:**
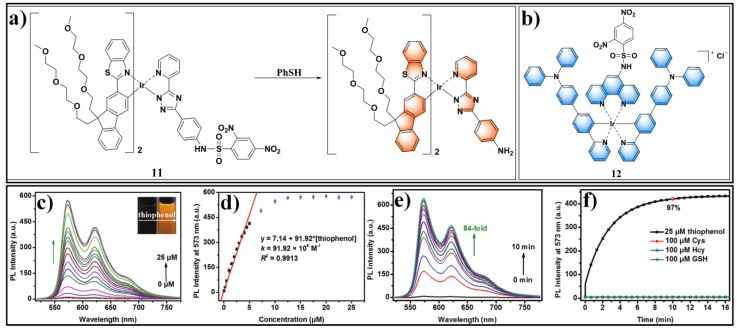
(**a**) Structure and reaction of probe **11** with PhSH. (**b**) Structure of probe **12**. (**c**) Phosphorescence spectra of probe **11** in the presence of different concentrations of thiophenol. (**d**) The correlation between phosphorescence intensity of probe **11** at 573 nm and concentration of thiophenol. (**e**) Phosphorescence spectra of probe **11** in the presence of thiophenol at different reaction times. (**f**) Fluorescence enhancement profile of phosphorescence intensity at 573 nm of probe **11** in the presence of thiophenol (five equivalents) or thiol species (20 equivalents) under nitrogen conditions. Reproduced with permission from Reference [71]; copyright Royal Society of Chemistry.

**Figure 6 molecules-24-03716-f006:**
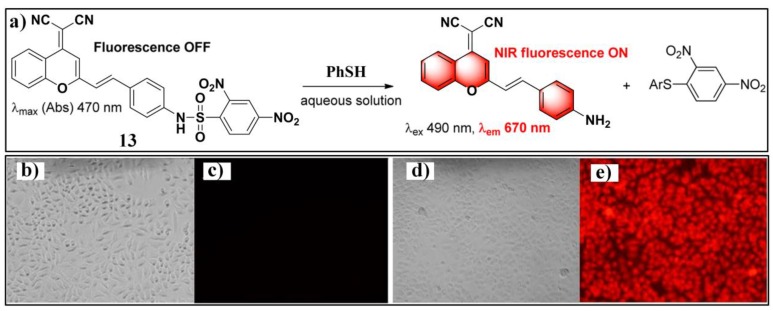
(**a**) Structure and reaction of probe **13** with PhSH. (**b**) Bright-field images of HeLa cells after being treated with probe **13**. (**d**) HeLa cells preincubated with thiophenol and then incubated with probe **13**. (**c**,**e**) Fluorescence images of (**b**,**d**), respectively. Reproduced with permission from Reference [76]; copyright American Chemical Society.

**Figure 7 molecules-24-03716-f007:**
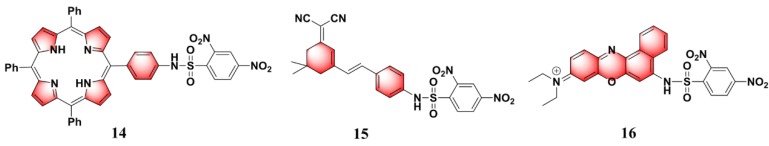
Other near-infrared (NIR) fluorescent PhSH probes (**14**–**16**) based on the recognition unit of 2,4-dinitrobenzenesulfonic amide.

**Figure 8 molecules-24-03716-f008:**
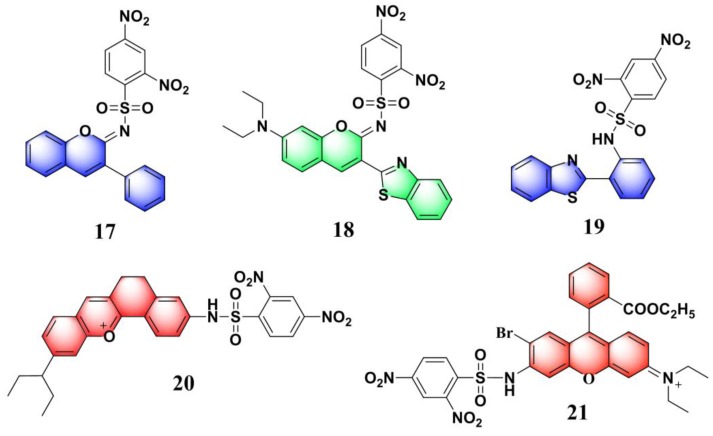
Fluorescent PhSH probes (**17**–**21**) based on the recognition unit of 2,4-dinitrobenzenesulfonic amide.

**Figure 9 molecules-24-03716-f009:**
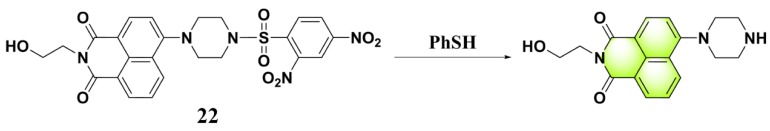
Structure and reaction of probe **22** with PhSH.

**Figure 10 molecules-24-03716-f010:**
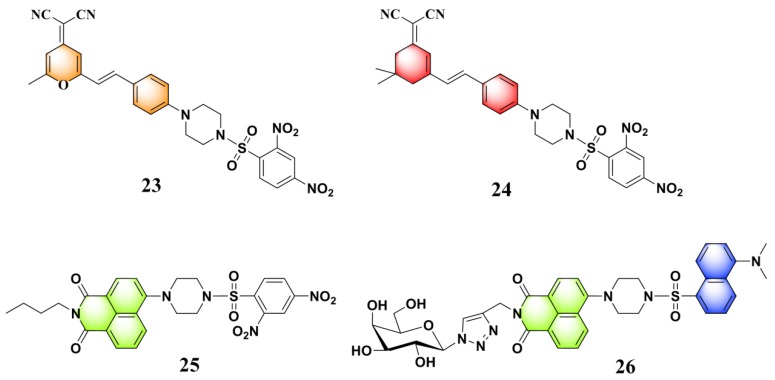
Fluorescent PhSH probes (**23**–**26**) containing the piperazine linker.

**Figure 11 molecules-24-03716-f011:**
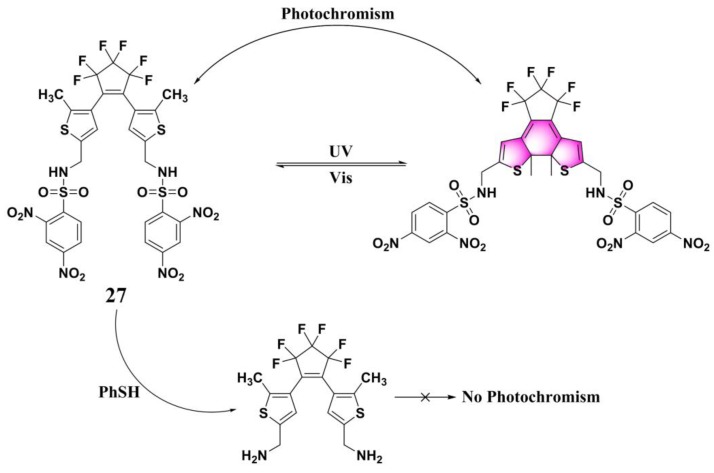
Structure and the sensing mechanism of probe **27** for PhSH.

**Figure 12 molecules-24-03716-f012:**
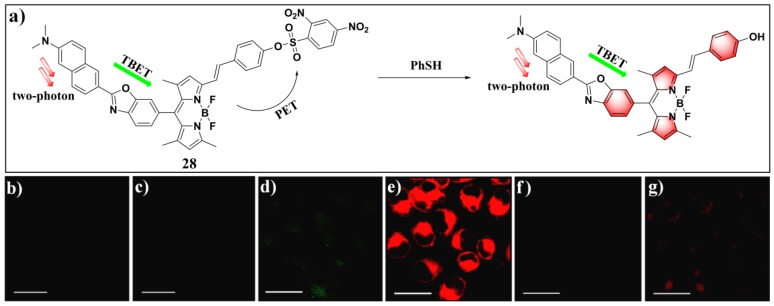
(**a**) Structure and reaction of probe **28** with PhSH. (**b**,**d**,**f**) Thymidine phosphorylase (TP) images of probe **28** in Hela cells from the green channel without thiophenol, with thiophenol, and with both NEM (N-ethylmaleimide) and thiophenol, respectively. (**c**,**e**,**g**) are corresponding TP images of (**b**,**d**,**f**) from the red channel, respectively. Reproduced with permissions from [89]. Copyright American Chemical Society.

**Figure 13 molecules-24-03716-f013:**
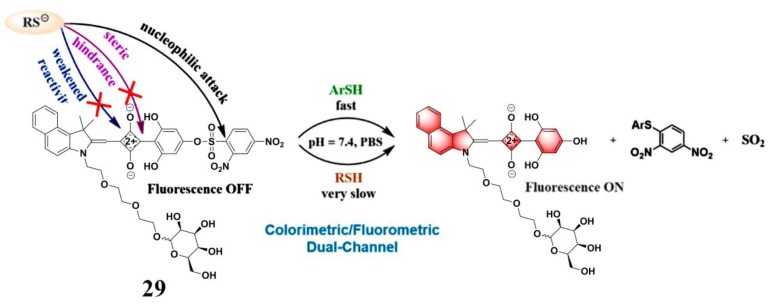
Structure and reaction of probe **29** with PhSH. Reproduced with permission from Reference [93]; copyright American Chemical Society.

**Figure 14 molecules-24-03716-f014:**
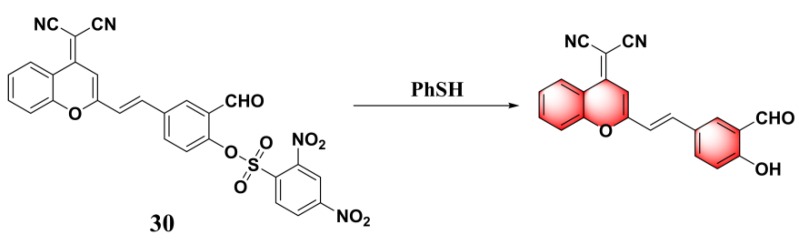
Structure and reaction of probe **30** with PhSH.

**Figure 15 molecules-24-03716-f015:**
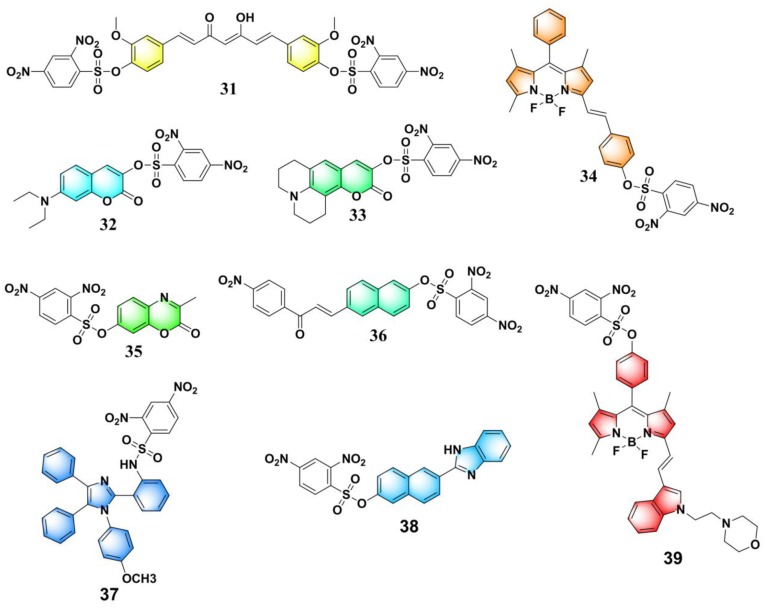
Fluorescent PhSH probes (**31**–**39**) based on the recognition unit of 2,4-dinitrobenzenesulfonate.

**Figure 16 molecules-24-03716-f016:**
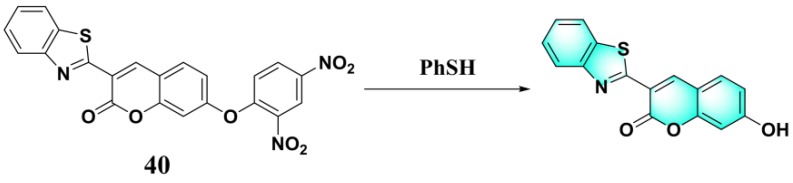
Structure and reaction of probe **40** with PhSH.

**Figure 17 molecules-24-03716-f017:**
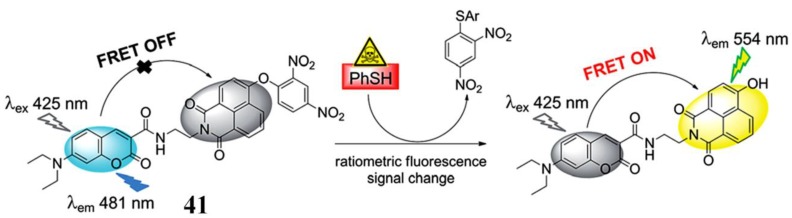
Structure and reaction of probe **41** with PhSH. Reproduced with permission from Reference [109]; copyright Royal Society of Chemistry.

**Figure 18 molecules-24-03716-f018:**
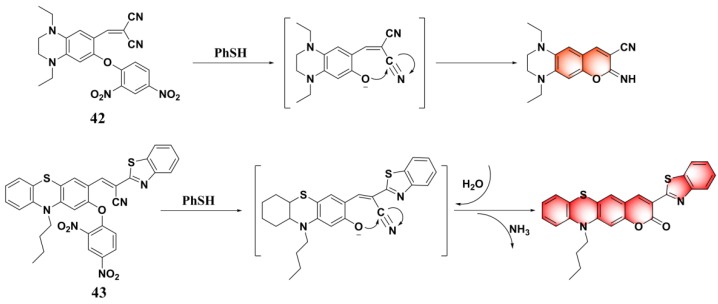
Structures and sensing mechanisms of probes **42**–**43** for PhSH.

**Figure 19 molecules-24-03716-f019:**
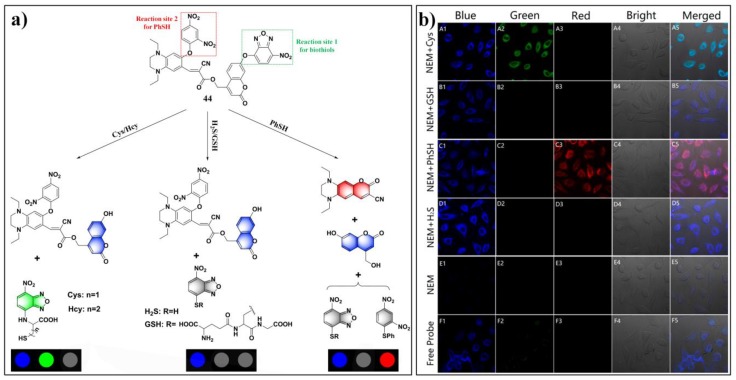
(**a**) Schematic illustration of the sensing mechanism of probe **44** for differentiating cysteine (Cys)/homocysteine (Hcy), glutathione (GSH)/H_2_S, and thiophenol. (**b**) Images of Hela cells incubated with different regents. Reproduced with permission from Reference [112]; copyright American Chemical Society.

**Figure 20 molecules-24-03716-f020:**
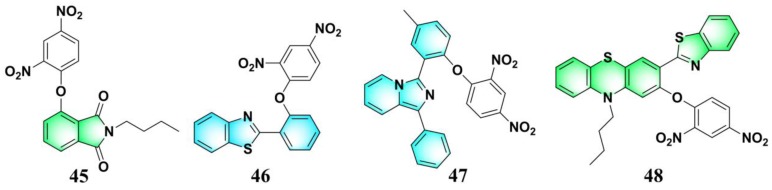
Fluorescent PhSH probes (**45**–**48**) based on excited-state intramolecular proton transfer (ESIPT) fluorophores.

**Figure 21 molecules-24-03716-f021:**
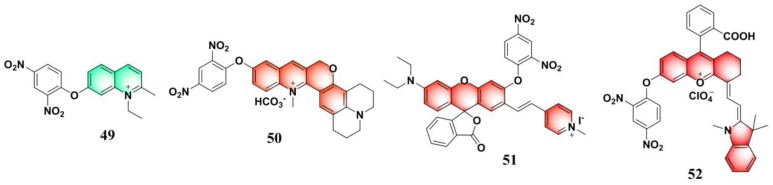
Fluorescent PhSH probes (**49–52**) based on positively charged fluorophores.

**Figure 22 molecules-24-03716-f022:**
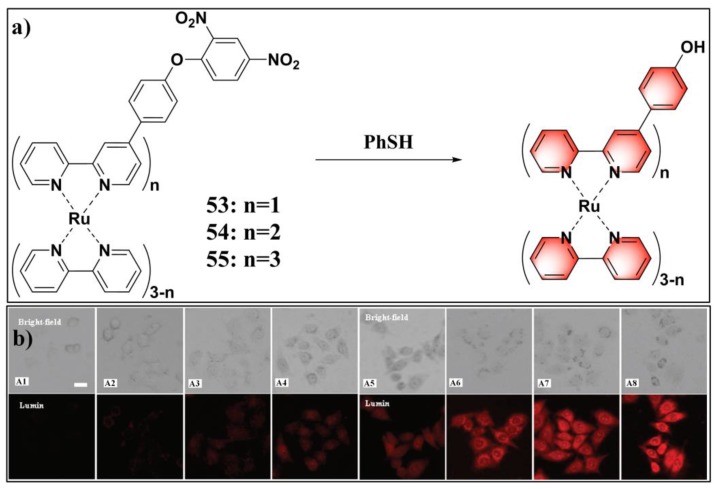
(**a**) Structure and reaction of probes **53**–**55** with PhSH. (**b**) Images of **54**-loaded Hela cells incubated with PhSH for different times, from A1 to A8 (0, 5, 10, 20, 30, 40, 50, 60 min); scale bar: 10 μm. Reproduced with permission from Reference [131]; copyright American Chemical Society.

**Figure 23 molecules-24-03716-f023:**
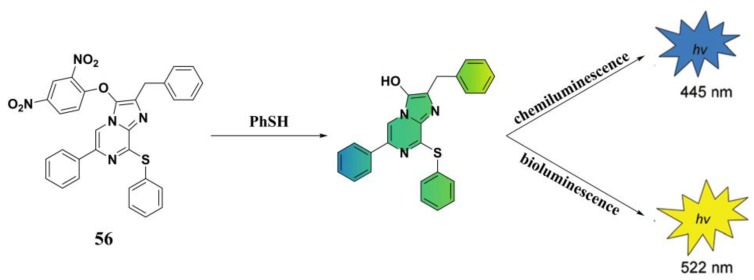
The chemiluminescent and bioluminescent sensing of probe **56** for PhSH.

**Figure 24 molecules-24-03716-f024:**
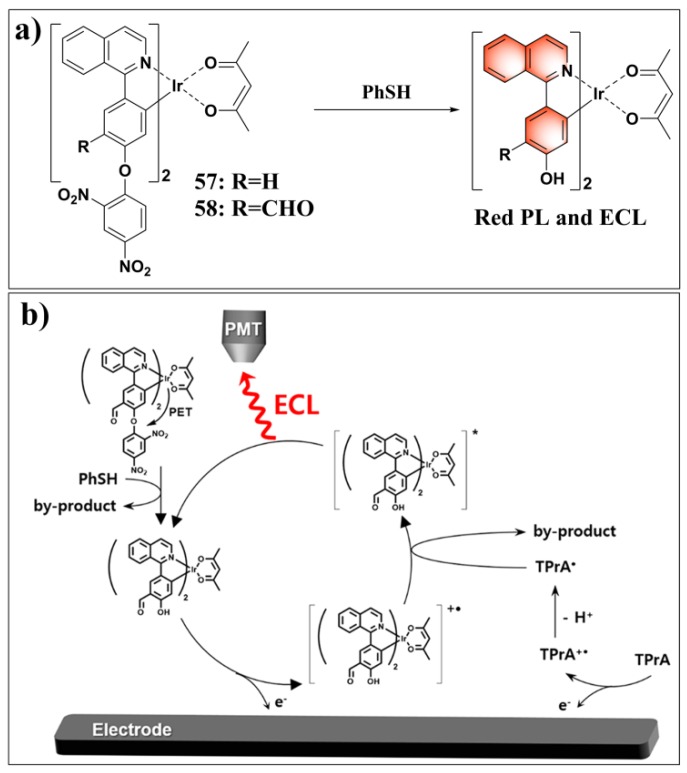
(**a**) Structures and reactions of probes **57**–**58** with PhSH. (**b**) Electrochemiluminescence (ECL) sensing mechanism of probe **58** for PhSH. Reproduced with permission from Reference [137]; copyright American Chemical Society.

**Figure 25 molecules-24-03716-f025:**
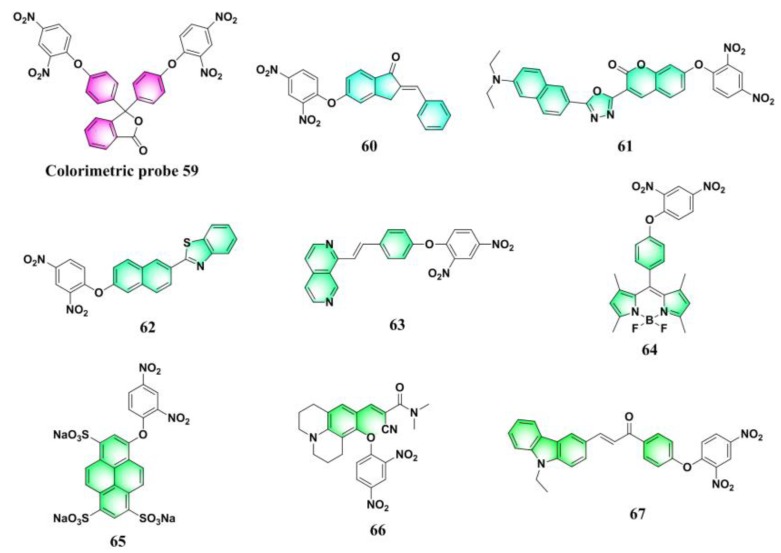
Structures of PhSH probes (**59**–**67**).

**Figure 26 molecules-24-03716-f026:**
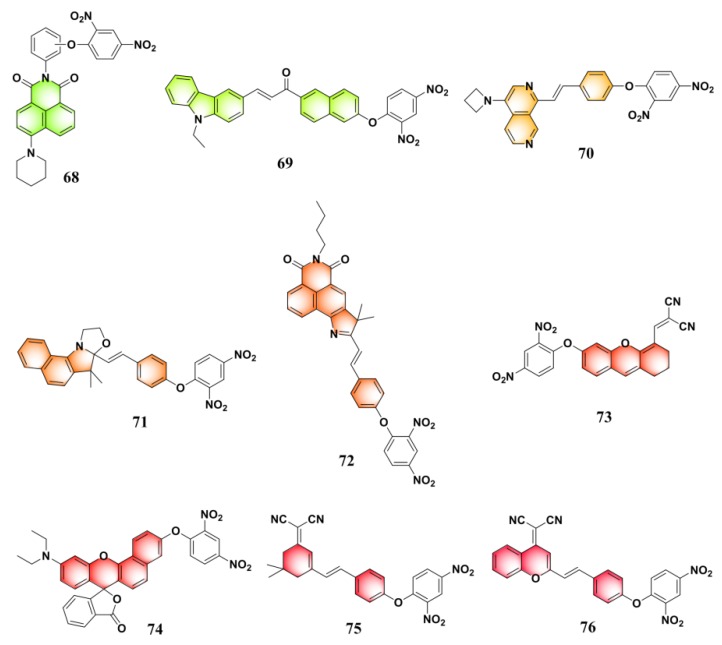
Structures of fluorescent PhSH probes (**68**–**76**).

**Figure 27 molecules-24-03716-f027:**
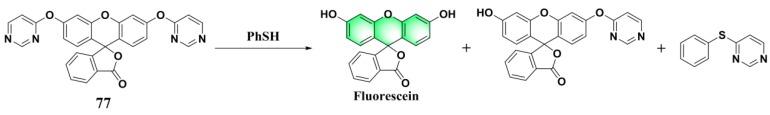
Structure and sensing mechanism of probe **77** for PhSH.

**Figure 28 molecules-24-03716-f028:**
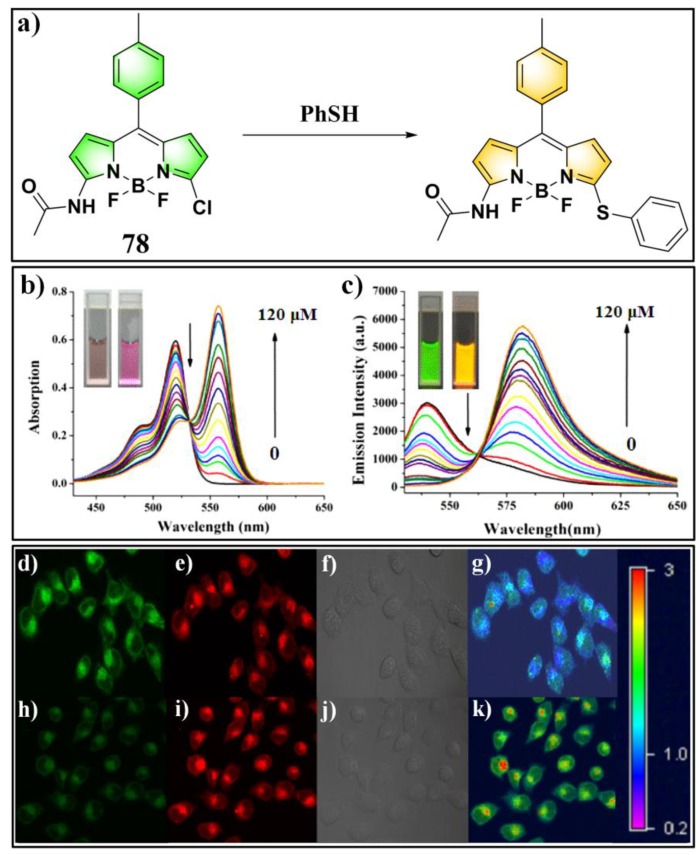
(**a**) Structure and reaction of probe **78** for PhSH. (**b**) Absorption and (**c**) emission spectra of **78** in the presence of PhSH with different concentrations. (**d**–**k**) Confocal fluorescence images of living HeLa cells: (**d**–**g**) cells loaded with probe **78**; (**h**–**k**) PhSH-incubated cells loaded with probe **78**; (**d**,**h**) green channel images (500–550 nm); (**e**,**i**) red channel images (570–620 nM); (**f**,**j**) bright-field images; (**g**) ratio image merged from (**d**,**e**); (**k**) ratio image merged from (**h**,**i**). Reproduced with permission from Reference [156]; copyright 2017 Elsevier B.V., New York, NY, USA.

**Figure 29 molecules-24-03716-f029:**
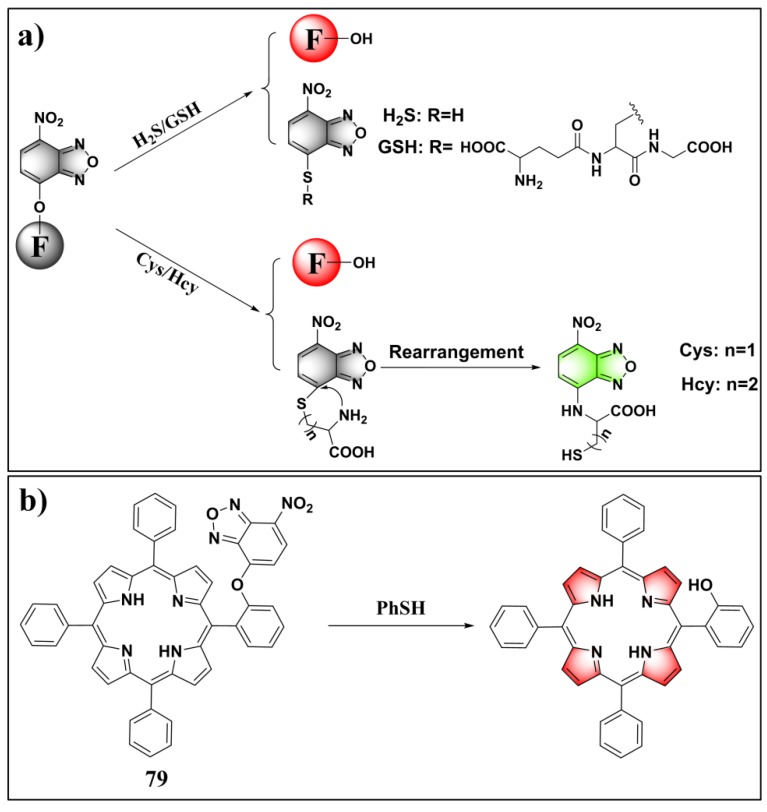
(**a**) Schematic illustration of nitrobenzoxadiazole (NBD)-appended dual-fluorophore system for differentiating H_2_S/GSH and Cys/Hcy. (**b**) Structure and reaction of probe **79** for PhSH.

**Table 1 molecules-24-03716-t001:** Comparison of fluorescent probes for thiophenol (PhSH).

Probes	λ_ex_/λ_em_ (nm)	Media	LOD	Response Time	Real Samples	Ref.
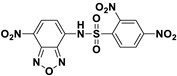	465/555	Phosphate buffer (pH 7.3, 0.01 M)	--	10 min	--	[51]
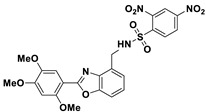	335/403	Phosphate buffer (pH 7.3, 0.01 M)	2 μM	20 min	--	[52]
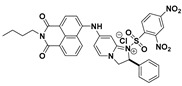	481/590	Phosphate buffer (pH 8.0, 0.01 M)	20 nM	15 min	River water	[53]
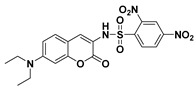	370/515	Phosphate buffer (pH 7.4, 0.01 M)	30 nM	~30 min	Lake water HEK293 cells	[54]
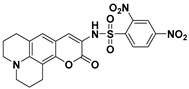	380/535	Phosphate buffer (pH 7.4, 0.01 M)	4.5 nM	~30 min	River water Lake water HEK293 cells	[55]
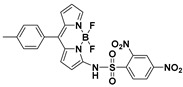	444/529	DMSO/PBS (1/99, pH 7.4, 0.01 M)	34.4 nM	10 min	HeLa cells	[60]
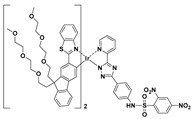	420/573	CH_3_OH/PBS (4/6, pH 7.4, 0.01 M)	2.5 nM	10 min	River water Lake water	[71]
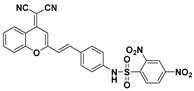	470/670	DMSO/PBS (3/7, pH 7.4, 0.01 M)	0.15 μM	10 min	River water Lake water HeLa cells	[76]
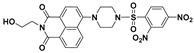	380/517	EtOH/H_2_O (7/3)	10.3 nM	10 min	Lake water Tap water Test paper	[84]
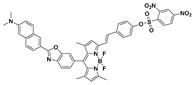	420/586	EtOH/PBS (1/1, pH 7.4, 0.01 M)	4.9 nM	20 min	River water HeLa cells Tissue slices	[89]
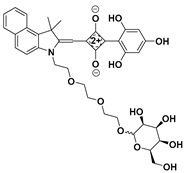	538/645	PBS (pH 7.4, 0.01 M)	9.9 nM	10 min	River water Lake water Tap water	[93]
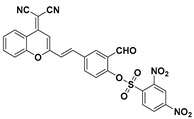	540/658	DMSO/PBS (1/1, pH 7.4, 0.01 M)	0.22 μM	3 min	Tap water Lake water HeLa cells	[94]
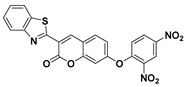	461/494	DMF/phosphate buffer (45/55, pH 7.0, 25 mM)	1.8 nM	30 min	Hela cells Soil sample Test paper	[104]
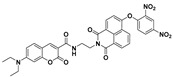	425/481, 554	DMF/PBS (4/6, pH 7.4, 50 mM)	0.12 μM	5 min	River water Lake water	[109]
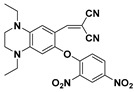	477/606	HEPES (pH 7.4, 20 mM, 1.0 mM CTAB)	8.2 nM	10 min	Tap water River water HeLa cells	[110]
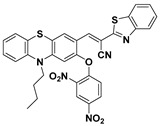	482/623	Phosphate buffer (pH 7.4, 20 mM, 1.0 mM CTAB)	2.9 nM	30 min	Tap water River water HeLa cells	[111]
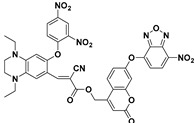	432/490, 624	CH_3_CN/PBS (3/7, pH 7.4, 0.01 M)	0.34 μM	20 min	HeLa cells	[112]
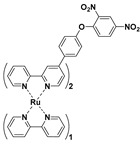	459/616	HEPES (pH 7.0, 20 mM)	32.9 nM	~30 min	HeLa cells	[131]
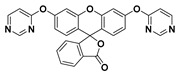	460/519	HEPES (pH 7.0, 20 mM)	1.8 μM	30 min	HepG2 cells	[155]
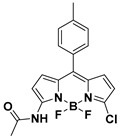	520/540, 581	CH_3_CN/HEPES (1/3, pH 7.4, 20 mM)	36.9 nM	~10 min	River water Lake water HeLa cells	[156]
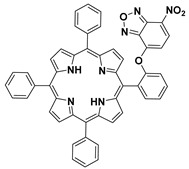	412/640	CH_3_CN/PBS (1/1, pH 7.4, 10 mM)	54 nM	60 s	--	[160]

LOD: limit of detection, DMSO: dimethyl sulfoxide, PBS: phosphate buffer saline, DMF: N,N-dimethylformamide, HEPES: N-2-hydroxyethylpiperazine-N-2′-ethanesulfonic acid, CTAB: cetyltrimethylammonium bromide

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
