# Peer review of "Recent Progress in the Development of Fluorescent Probes for Thiophenol"

_molecules, 2019, doi:10.3390/molecules24203716_

Round 1
Reviewer 1 Report
- It would be good to append/add the equation for the determination of the quantum yield used by the authors, since there are different ways of doing it and if these are different, the comparison between them for better effectiveness or a simpler way of calculating.
- In the conclusions, they cite the selective detection since 2007, but the references begin since 2005 by Yang et al because it was not considered from this.
- All molecules are synthetic for detection or biological use there are studies of biocompatibility.
- For the first compound structures if fluorescence spectra were attached, it would help to make comparisons since only one increase or decrease is mentioned in Figure 4 and 5 the structures are different and both show strong fluorescence.
- A comparative table could be made showing the structures of the probes used comparing their structural characteristics / applications / the effect of the fluorophore used. This table would help the classification that the authors mention in the abstract.
Author Response
Comment 1: It would be good to append/add the equation for the determination of the quantum yield used by the authors, since there are different ways of doing it and if these are different, the comparison between them for better effectiveness or a simpler way of calculating.
Response: Thanks very much for the reviewer’s constructive suggestion, in the revised manuscript, we have presented the methods for quantum yield calculations according to the cited references.
On page 2, lines 67-68, “…quantum yield was determined by reference to harmine in 0.1 N H2SO4 (Φ’ = 0.45)…”
On page 9, lines 241-246, “The quantum yield was calculated using the equation of Φs = Φr (ArFs/AsFr) (ns/nr)2, where As and Ar are the absorbance of the sample and the reference at the excitation wavelength, respectively, Fs and Fr are the corresponding relative integrated fluorescence intensities, and n is the refractive index of the solvent, Φs and Φr are the quantum yields of the sample and the reference, respectively, quinine sulfate (Φr = 0.546 in 1N H2SO4) used as the reference.”
Comment 2: In the conclusions, they cite the selective detection since 2007, but the references begin since 2005 by Yang et al because it was not considered from this.
Response: The first fluorescent probe for the selective detection of PhSH was developed by Wang et. al. (Jiang, W.; Fu, Q.; Fan, H.; Ho, J.; Wang, W. A Highly Selective Fluorescent Probe for Thiophenols. Angew. Chem. Int. Ed. 2007, 46, 8445-8448.).
In the manuscript, we firstly introduced the properties and applications of the targe thiophenol, so we cited a relavent work (Yang, L.; Feng, J.-K.; Liao, Y.; Ren, A.-M. A theoretical investigation on the electronic and optical properties of π-conjugated copolymers with an efficient electron-accepting unit bithieno[3,2-b:2′3′-e]pyridine. Polymer 2005, 46, 9955-9964.),
Comment 3: All molecules are synthetic for detection or biological use there are studies of biocompatibility.
Response: Most of these reported fluorescent PhSH probes are synthetic organic molecules. Biocompatibility is very important especially for biological applications. Some of these reported probes have been exploited for imaging PhSH in living cells, and accordingly their biocompatibility have been evaluated by cytotoxicity studies. For example, on page 12, lines 336-338, “Furthermore, the probe exhibited excellent biocompatibility and cell-membrane permeability, and has been successfully exploited for imaging PhSH in living HeLa cells as well as in living mice.”; page 13, lines 356-358, “Further cellular imaging experiments demonstrated that probe 54 has excellent intracellular retention, long-term stability to resist photobleaching, low cytotoxicity, and the tendency of nuclear localization.”.
Comment 4: For the first compound structures if fluorescence spectra were attached, it would help to make comparisons since only one increase or decrease is mentioned in Figure 4 and 5 the structures are different and both show strong fluorescence.
Response: According to the reviewer’s suggestion, we have attached the spectral changes of the first probe in Figure 4 and in Figure 5.
Figure 4. (a) Fluorescent PhSH probes (6-10) based on boron-containing dyes. (b) Reaction of probe 6 with PhSH. (c) energy level diagram of the frontier molecular orbitals (MOs) of probe 6 and corresponding amine. (d) UV-vis absorption spectra of probe 6 before and after addition of PhSH, photographs taken under ambient light. (e) Fluorescence spectra of 6 in the presence of different concentrations of PhSH. Photographs taken under a hand-held UV lamp.
Figure 5. (a) Structure and reaction of probe 11 with PhSH. (b) Structure of probe 12. (c) Phosphorescence spectra of probe 11 in the presence of different concentrations of thiophenol. (d) the correlation between phosphorescence intensity of probe 11 at 573 nm and concentration of thiophenol. (e) Phosphorescence spectra of probe 11 in the presence of thiophenol at different reaction time. (f) Fluorescence enhancement profile of phosphorescence intensity at 573 nm of probe 11 in the presence of thiophenol (5 equiv.) or thiol species (20 equiv.) under nitrogen conditions.
Comment 5: A comparative table could be made showing the structures of the probes used comparing their structural characteristics / applications / the effect of the fluorophore used. This table would help the classification that the authors mention in the abstract.
Response: According to the reviewer’s suggestion, we have presented a comparative table showing the structures and analytical performances of some representative probes for sensing PhSH.
Table 1. Comparison of fluorescent probes for PhSH.
|
Probes |
λex/λem (nm) |
Media |
LOD
|
Response time |
Real samples |
Ref. |
|
465/555 |
phosphate buffer (pH 7.3, 0.01 M) |
-- |
10 min |
-- |
[51] |
|
|
335/403 |
phosphate buffer (pH 7.3, 0.01 M) |
2 μM |
20 min |
-- |
[52] |
|
|
481/590 |
phosphate buffer (pH 8.0, 0.01 M) |
20 nM |
15 min |
River water |
[53] |
|
|
370/515 |
phosphate buffer (pH 7.4, 0.01 M) |
30 nM |
~ 30 min |
Laker water HEK293 cells |
[54] |
|
|
380/535 |
phosphate buffer (pH 7.4, 0.01 M) |
4.5 nM |
~ 30 min |
River water Laker water HEK293 cells |
[55] |
|
|
444/529 |
DMSO/PBS (1/99, pH 7.4, 0.01 M) |
34.4 nM |
10 min |
HeLa cells |
[60] |
|
|
420/573 |
CH3OH/PBS (4/6, pH 7.4, 0.01 M) |
2.5 nM |
10 min |
River water Laker water
|
[71] |
|
|
470/670 |
DMSO/PBS (3/7, pH 7.4, 0.01 M) |
0.15 μM |
10 min |
River water Laker water HeLa cells |
[76] |
|
|
380/517 |
EtOH/H2O (7/3) |
10.3 nM |
10 min |
Laker water Tap water Test paper |
[84] |
|
|
420/586 |
EtOH/PBS (1/1, pH 7.4, 0.01 M) |
4.9 nM |
20 min |
River water HeLa cells Tissue slices |
[89] |
|
|
538/645 |
PBS (pH 7.4, 0.01 M) |
9.9 nM |
10 min |
River water Laker water Tap water
|
[93] |
|
|
540/658 |
DMSO/PBS (1/1, pH 7.4, 0.01 M) |
0.22 μM |
3 min |
Tape water Laker water HeLa cells |
[94] |
|
|
461/494 |
DMF/ phosphate buffer (45/55, pH 7.0, 25 mM) |
1.8 nM |
30 min |
Hela cells Soil sample Test paper |
[104] |
|
|
425/ 481,554 |
DMF/PBS (4/6, pH 7.4, 50 mM) |
0.12 μM |
5 min |
River water Laker water |
[109] |
|
|
477/606 |
HEPES (pH 7.4, 20 mM, 1.0 mM CTAB) |
8.2 nM |
10 min |
Tape water River water HeLa cells |
[110] |
|
|
482/623 |
phosphate buffer (pH 7.4, 20 mM, 1.0 mM CTAB) |
2.9 nM |
30 min |
Tape water River water HeLa cells |
[111] |
|
|
432/ 490,624 |
CH3CN/PBS (3/7, pH 7.4, 0.01 M) |
0.34 μM |
20 min |
HeLa cells |
[112] |
|
|
459/616 |
HEPES (pH 7.0, 20 mM) |
32.9 nM |
~ 30 min |
HeLa cells |
[131] |
|
|
460/519 |
HEPES (pH 7.0, 20 mM) |
1.8 μM |
30 min |
HepG2 cells |
[155] |
|
|
520/ 540,581 |
CH3CN/HEPES (1/3, pH 7.4, 20 mM) |
36.9 nM |
~ 10 min |
River water Laker water HeLa cells |
[156] |
|
|
412/640 |
CH3CN/PNS (1/1, pH 7.4, 10 mM) |
54 nM |
60 s |
-- |
[160] |

Reviewer 2 Report
In this manuscript, the authors presented a comprehensive review of recent examples of fluorescent probes for PhSH. The review is well organized and schemes are well drawn to show the important probe structure and its working mechanism. The references cited are all new reports published within the last decade. I would recommend acceptance of this manuscript for publication
Minor revision:
Line 46, "higher pKa" would be "lower pKa".
Author Response
Comment 1: Line 46, "higher pKa" would be "lower pKa".
Response: Thanks very much for the reviewer’s positive and encouraging comments. And "higher pKa" has been revised to "lower pKa".
